# Prevalence and risk factors of metabolic syndrome in Ethiopia: describing an emerging outbreak in HIV clinics of the sub-Saharan Africa – a cross-sectional study

Abdurezak Ahmed Abdela [1], Helen Yifter,[1] Ahmed Reja,[1] Aster Shewaamare,[2] Ighovwerha Ofotokun,[3] Wondwossen Amogne Degu[1]

¹Department of Internal Medicine, Addis Ababa University, Addis Ababa, Ethiopia
²Zewditu Memorial Hospital, Addis Ababa, Ethiopia
³Department of Internal Medicine, Emory University, Atlanta, Georgia, USA

**Correspondence to**
Dr Wondwossen Amogne Degu;
wonamogne@yahoo.com

## ABSTRACT

**Objectives** HIV-induced chronic inflammation, immune activation and combination antiretroviral therapy (cART) are linked with adverse metabolic changes known to cause cardiovascular adversities. This study evaluates the prevalence of lipodystrophy, and metabolic syndrome (MetS), and analyses risk factors in HIV-infected Ethiopians taking cART.

**Methods** A multicentre cross-sectional study was conducted at tertiary-level hospitals. Eligible participants attending the HIV clinics were enrolled. Sociodemographic, anthropometric, clinical, HIV treatment variables, lipid profile, fasting blood glucose level, risk factors and components of MetS, also lipodystrophy, were studied. Data were analysed by SPSS statistical package V.25 with descriptive and analytical statistics. For multivariable analysis of risk factors, a logistic regression model was used. Results were presented in frequency and percentages, mean±SD, or median+IQR. Statistical significance was taken as p<0.05.

**Results** Among 518 studied participants, two-thirds were females, and the mean age of the study population was 45 years (SD=11). The mean duration of cART was 10 years (SD=4). Median CD4 count was 460 cells/mm³. The prevalence of MetS according to the Adult Treatment Panel III (2005) criteria was 37.6%. In multivariable analysis, independent risk factors for MetS were age >45 years (aHR 1.8, 95% CI 1.2 to 2.4), female sex (aHR 1.8, 95% CI 1.1 to 2.8), body mass index (BMI)≥25 kg/m² (aHR 2.7, 95% CI 1.8 to 4.1), efavirenz-based cART (aHR 2.8, 95% CI 1.6 to 4.8) and lopinavir/ritonavir-based cART (aHR 3.7, 95% CI 1.0 to 13.3). The prevalence of lipodystrophy was 23.6%. Prior exposure to a stavudine-containing regimen was independently associated with lipodystrophy (aHR 3.1, 95% CI 1.6 to 6.1).

**Conclusion** Our study revealed 38% of the participants had MetS indicating considerable cardiovascular disease (CVD) risks. Independent risk factors for MetS were BMI≥25 kg/m², efavirenz and lopinavir/ritonavir-based cART, female sex and age ≥45 years. In addition to prevention, CVD risk stratification and management will reduce morbidity and mortality in people with HIV infection.

## STRENGTHS AND LIMITATIONS OF THIS STUDY

⇒ This study examined the cardiovascular risk factors and metabolic syndrome (MetS) among people with HIV infection (PWH) on combination antiretroviral therapy (cART) in a sub-Saharan African setting.

⇒ This study has the largest sample size compared with other studies done in Ethiopia about MetS and lipodystrophy in PWH on cART.

⇒ Complete nutritional assessment is lacking.

⇒ Advanced techniques for the assessment of lipodystrophy were not used.

## INTRODUCTION

HIV/AIDS is a global pandemic. There were an estimated 38.4 million individuals living with HIV infection in 2021.[1] More than 95% of the world's population with HIV reside in low-income and middle-income countries,[2] and two-thirds are in sub-Saharan Africa.[1] As the number of people receiving combination antiretroviral therapy (cART) progressed to 28.7 million in 2021,[1] AIDS-related mortality has decreased by half compared with 2010.[3] Such a phenomenal decrease in AIDS-related mortality is mainly attributed to the reduction of opportunistic infections.

People with HIV infection (PWH) receiving virus-suppressive cART will have immune reconstitution. Consequently, their life expectancy has become similar to HIV-uninfected individuals.[4] The data collection of adverse events of anti-HIV Drugs study revealed that 289 of 2482 (11.6%) deaths were related to cardiovascular disease (CVD).[5] Though this analysis was made for data until 2008, DAD data were obtained in cohorts I–III in the period of December 1999–2001, then throughout spring of 2004, and until 2010, respectively.

There is a shift from AIDS-related mortality to non-AIDS-related mortality as shown by a 2006 ART cohort collaboration (ART-CC). The study indicated that the frequency of opportunistic infection as a cause of death in PWH decreased from 32% to 19%. In contrast, mortality from CVD causes and type 2 diabetes mellitus complications increased from 1.3% to 4.3%.[6 7] CVD has become a particular concern because of cART-induced metabolic changes, the high prevalence of cardiovascular risk factors in PWH, and growing evidence of HIV-accelerated inflammatory processes that promote atherosclerosis.[8] As a consequence of some antiretroviral drugs changing the body's morphology because of fat redistribution and causing dyslipidaemia, there is an increased risk for cardiovascular morbidity and mortality in PWH.[9] Metabolic syndrome (MetS), a combination of known cardiovascular risk factors and HIV-associated lipodystrophy have been reported in different countries.[10 11] In PWH, MetS was detected in 21.1%, and HIV-associated lipodystrophy was detected in 12.1%.[12] Hypertriglyceredimaeia prevalence was 15.2%.[13] Individual components of MetS and metabolic changes with lipodystrophy are known to be risk factors for CVD.

Identifying CVD risk factors in PWH on cART has public health significance. We evaluated the prevalence of MetS, lipodystrophy and related risk factors among Ethiopian PWH on cART at two tertiary-level hospitals in Addis Ababa, Ethiopia.

## METHODS
### Study design and eligibility
A cross-sectional study was conducted at tertiary referral hospitals, namely Tikur Anbessa Specialized Hospital (TASH) and Zewditu Memorial Hospital (ZMH), in Addis Ababa, Ethiopia. TASH is Addis Ababa University's main teaching hospital and the Addis Ababa Health Bureau administers ZMH. Addis Ababa has 2.7 million people accounting for nearly 4% of the Ethiopian population.[14] TASH and ZMH have a total of 4738 and 7200 HIV patients on cART at the time of the study, respectively. In these institutions, the largest federal government-supported HIV/AIDS care and treatment programme including the provision of cART and opportunistic infections prophylaxis and treatment is taking place. The study was conducted from July 2018 to June 2019. The inclusion criteria were PWH, age 18 and above, on cART and adherent to their treatment (adherence to their treatment for ≥95% (≤3 doses missed out of 60 doses)) and taking cART for at least 12 months.[12 13]

The exclusion criteria were: non-adherence to cART, diagnosis of type 2 diabetes mellitus or hypertension before cART initiation, hypothyroidism, chronic renal failure, chronic corticosteroids use, oral contraceptives, pregnancy, unable to undergo weight, height and abdominal circumference measurements, unwilling to provide a blood sample for analysis and subjects who are too sick to be interviewed and examined.

The study subjects were screened by reviewing their health records and with interviews using structured questionnaires. A total of 518 subjects were recruited for the study using a simple random sampling method.

### Patient and public involvement
There was no patient or public involvement in the study design, or conduct, or reporting, or dissemination plans of our study findings.

### Data collection and quality assurance
Data were collected by trained and qualified HIV specialty clinic nurses through interviews about sociodemographic characteristics such as age, sex, educational status, income and occupation. The patient's health record was reviewed for CD4 counts and HIV RNA levels measured in the last 6 months before study enrolment. Data about the duration of HIV and the course of treatment with cART were examined. The data include the duration of treatment with the current and changed regimen, the specific regimens, regimen changes, the reason for the change and the total duration of treatment with cART regimens was collected. The structured questionnaire also contained rest of clinical questions. The patient's self-report during interview and physical examination were used to collect body fat redistribution data of lipodystrophy manifestations including loss of buccal fat pad, central adiposity with/without extremity wasting, dorsocervical/supraclavicular fat pad and breast enlargement that were documented during the patients' interview and physical examination. The height and body weight were measured in centimetres and kilograms, respectively, using a digital balance with height measurement attached to it. The height measurements were converted into metres. Body mass index (BMI) was calculated as a ratio between the weight in kilograms and the square of the height in metres.[15] Abdominal waist circumference measurement was taken at the approximate midpoint between the lower margin of the last palpable rib and the top of the iliac crest as described in WHO guidelines.[16]

Using an automated sphygmomanometer, resting blood pressure in mm Hg was measured based on standard definitions.[17] Morning Fasting venous blood sample, 5 mL, was collected after an overnight, minimum of 8 hours, fasting for fasting blood sugar (FBS), total cholestrol (TC), triglyceride (TG), high density lipoprotien-cholestrol (HDL-C) and low-density lipoprotien-cholestrol (LDL-C) levels determination. Blood samples were collected in Gel and clot activator vacutainer tubes (Henso Medical (Hangzhou)), taken to the lab in the same morning, kept at room temperature and analysed within the same day. Laboratory tests were made for FBS, TC, TG, HDL-C and LDL-C by direct measurement using reagents (Human Diagnostics worldwide LiquiColor, Germany-Wiesbaden) for all by Mindray chemistry machine. Data were checked for accuracy, consistency and cleanliness.

**Table 1** Sociodemographic and clinical characteristics of study participants at TASH and ZMH, Addis Ababa, Ethiopia

| Sociodemographic and clinical variables | Frequency | % |
|---|---|---|
| Sex | | |
| Male | 170 | 32.8 |
| Female | 347 | 67.0 |
| Education | | |
| No formal education attended | 48 | 9.30 |
| Primary and secondary school | 386 | 74.5 |
| Diploma and higher degree | 84 | 16.2 |
| Occupation | | |
| Private works | 227 | 43.8 |
| Civil servant | 107 | 20.7 |
| Merchant | 28 | 5.40 |
| Housewife | 69 | 13.3 |
| Miscellaneous others | 87 | 16.8 |
| Types of current cART* | | |
| (1d, 1c, 1g, 1h, 1f) | 158 | 30.5 |
| 1e | 297 | 57.3 |
| (2g, 2h, 2i, 2f) | 63 | 12.2 |
| Initial cART changed | | |
| Yes | 213 | 41.1 |
| No | 305 | 58.9 |
| Types of changed initial cART* | | |
| (1a,1b) | 69 | 32.4 |
| (1c,1d) | 91 | 42.7 |
| (1e,1g,1f) | 32 | 15.0 |
| (2i,2f,2h) | 16 | 7.50 |
| Missing-not traceable | 05 | 2.40 |
| Reasons for change of cART* | | |
| Treatment failure | 68 | 31.9 |
| Lipodystrophy | 75 | 35.2 |
| Anaemia | 26 | 12.2 |
| Others | 30 | 14.1 |
| Missing-not traceable | 14 | 6.60 |
| | **Mean±SD** | **95% CI for mean** |
| Age | 45.3±10.74 | 44.37 to 46.20 |
| Duration of | | |
| HIV since diagnosed | 10.8±3.85 | 10.46 to 11.12 |
| Treatment with cART | 9.97±3.63 | 9.65 to 10.27 |
| Treatment with current cART | 7.90±3.79 | 7.51 to 08.19 |
| Treatment with changed cART | 5.10±3.10 | 4.68 to 05.58 |

Continued

**Table 1** Continued

| Sociodemographic and clinical variables | Frequency | % |
|---|---|---|
| BMI (kg/m$^2$) | 24.6±5.02 | 23.97 to 25.40 |
| Waist circumference (cm) | 90.6±9.48 | 89.09 to 91.90 |
| Blood pressure (mm Hg) | | |
| Systolic | 122±15.38 | 119.9 to 124.3 |
| Diastolic | 79.5±10.9 | 77.96 to 81.05 |
| | **Median** | **IQR** |
| Recent CD4 count (cells/mm$^3$) | 460 | (326.5 to 633.5) |

*cART: combined antiretroviral therapy;1a: stavudine (D4T)/lamuvidin (3TC)/nevirapine (NVP); 1b: D4T/3TC/efavirenz (EFV); 1c: zidovudine (ZDV)/3TC/NVP; 1d: ZDV/3TC/EFV; 1e: tenofovir (TDF)/3TC/EFV; 1 f:TDF/3TC/NVP; 1g: abacavir (ABC)/3TC/EFV;1 hour: ABC/3TC/NVP; 2e: AZT/3TC/ritonavir boosted lopinavir (LPV/r); 2f: AZT/3TC/ritonavir boosted atazanavir (ATV/r); 2g: TDF/3TC/(LPV/r); 2 hours: TDF/3TC/ATV/r;2i: ABC/3TC/LPV/r.
BMI, body mass index; cART, combination antiretroviral therapy; TASH, Tikur Anbessa Specialized Hospital; ZMH, Zewditu Memorial Hospital.

### Assessment of MetS and lipodystrophy

MetS was defined using International Diabetes Federation (IDF) and the National Cholesterol Education Program's Adult Treatment Panel (ATPIII) criteria.[18 19] IDF criteria are waist circumference ≥80 cm in women and ≥94 cm in men plus two of the following: TG ≥150 mg/dL or drug treatment for high TG, HDL<40 mg/dL in men or <50 mg/dL in women or drug treatment for low HDL-C, FBS ≥100 mg/dL or treatment for diagnosed diabetes, or blood pressure ≥130/85 mm Hg or drug treatment for hypertension. ATPIII criteria are three of the following: waist circumference ≥88 cm in women and ≥102 cm in men, TG ≥150 mg/dL or drug treatment for elevated TG, HDL<40 mg/dL in men or <50 mg/dL in women or drug treatment for low HDL-C, FBS≥100 mg/dL or drug treatment for elevated blood glucose, or blood pressure ≥130/85 mm Hg or drug treatment for hypertension. Dyslipidaemia was defined by ATP III criteria with TC<200 mg/dL as desirable, TG<150 mg/dL as normal, HDL-C<40 mg/dL and <50 mg/dL for men and women, respectively, as low, and LDL-C<100 mg/dL as optimal.[19] Lipodystrophy was defined as having any of clinical manifestations, self-reported or on physical examination, of loss of buccal fat pad, abdominal obesity with/without extremity wasting, breast enlargement or dorsocervical or supraclavicular fat pad accumulation with or without signs of insulin resistance.[20–23]

### Statistical analysis

Data were entered into an Excel sheet, cleaned and analysed with SPSS statistical package V.25. Descriptive statistical analysis was made for sociodemographic and clinical variables and reported as frequency and percent,

mean±SD or median+IQR. The prevalence of MetS and lipodystrophy was computed and reported as percentages. Bivariate analysis was used to identify significant predictors of MetS and lipodystrophy. Independent variables showing association during bivariate analysis were fed into multivariate models to control for confounding variables and analysed using multivariable logistic regression. Because only a few study participants had the habit of smoking this factor was not tested for association with dependent variables. All significant test values were two sided, and the significance level was set at $p<0.05$.

## RESULTS

### Patients characteristics

A total of 518 participants were enrolled, men (170 (32.8%)) and women (347 (67.0%)), and their mean (SD) age was 45.3 (10.7) years. Most of the study participants were private workers (44%), followed by civil servants (21%). Nearly 60% of the participants took a fixed-formulation tablet consisting of tenofovir, lamivudine and efavirenz during the study (table 1). In general, 213 (41.1%) of the participants had their first-line cART regimen subsequently changed. The changed regimens consisted of stavudine, lamivudine, efavirenz or nevirapine (69 (32.4%)); and zidovudine, lamivudine, efavirenz or nevirapine (91 (42.7%)).

The most common reasons for the changes were HIV treatment failure (68 (31.9%)), lipodystrophy (75 (35.2%)) and anaemia(26 (12.2%)).The mean(SD) duration of HIV diagnosis, total duration of cART, treatment with current cART and treatment with initially changed cART in years were 10.8 (3.85), 9.97 (3.63), 7.90 (3.79) and 5.1 (3.10), respectively. The mean BMI (SD) was 24.6 (5.02) kg/m$^2$ and the mean abdominal waist circumference (SD) was 90.6 (9.5) cm. The median CD4 cell counts of the study population were 460 (IQR 326.5–633.5) cells/mm$^3$ (table 1).

### Prevalence of MetS, lipodystrophy, hypertriglyceridaemia and distribution of lipodystrophy manifestations

In this cohort of 518 study participants, the prevalence of MetS by IDF 2006 criteria was found to be 193 (37.3%) and by ATPIII 2005 criteria 195 (37.6%). The prevalence of lipodystrophy was 23.6% (122 cases). The prevalence of hypertriglycridaemia was 40.7% (211 cases) (table 2). Central adiposity±extremity wasting was the highest 80 (15.4%) participants reported lipodystrophy manifestation followed by loss of buccal fat pad 76 (14.7%). The highest lipodystrophy manifestation by physical exam was central adiposity±extremity wasting 61 (11.8%), followed by loss of buccal fat pad 60 (11.6%) (table 2).

### Associated risk factors for MetS and lipodystrophy

Bivariate regression analysis revealed a significant association with MetS between age ≥45 years (cHR 1.68, 95% CI 1.1 to 2.5, p=0.009), female sex (cHR 1.71, 95% CI 1.2 to 2.5, p=0.003), BMI (kg/m$^2$) ≥25 (cHR 3.1, 95% CI 2.1 to

**Table 2** Prevalence of metabolic syndrome, lipodystrophy, hypertriglyceridaemia and the distribution of lipodystrophy manifestations in HIV patients on cART at TASH and ZMH, Addis Ababa, Ethiopia

| Variables | Frequency | % |
|---|---|---|
| Metabolic syndrome IDF 2006 | | |
| Yes | 193 | 37.3 |
| No | 325 | 62.7 |
| Metabolic syndrome ATP III 2005 | | |
| Yes | 195 | 37.6 |
| No | 323 | 62.4 |
| Lipodystrophy | | |
| Participants reported | | |
| Yes | 117 | 22.6 |
| No | 401 | 77.4 |
| Physical examination | | |
| Yes | 122 | 23.6 |
| No | 396 | 76.4 |
| Hypertrigleceridaemia | | |
| Yes | 211 | 40.7 |
| No | 307 | 59.3 |
| Lipodystrophy manifestations: participants reported | | |
| Loss of buccal fat pad | | |
| Yes | 76 | 14.7 |
| No | 442 | 85.3 |
| Central adiposity±extremity wasting | | |
| Yes | 80 | 15.4 |
| No | 438 | 84.6 |
| Supraclavicular/dorsocervical fat pad | | |
| Yes | 39 | 7.5 |
| No | 479 | 92.5 |
| Breast enlargement | | |
| Yes | 31 | 6.0 |
| No | 486 | 93.8 |
| Lipodystrophy manifestations: physical examination | | |
| Loss of buccal fat pad | 60 | 11.6 |
| Central adiposity±extremity wasting | 61 | 11.8 |
| Supraclavicular/dorsocervical fat pad | 1 | 0.2 |
| No sign of lipodystrophy | 396 | 76.4 |

ATP III, Adult Treatment Panel; cART, combined antiretroviral treatment; IDF, International Diabetes Federation; TASH, Tikur Anbessa Specialized Hospital; ZMH, Zewditu Memorial Hospital.

4.4), p=0.001) and efavirenz (EFV)-based cART regimen (cHR 2.8, 95% CI 1.71 to 4.65, p=0.001) based on IDF 2006 criteria.

In contrast to the nevirapine (NVP)-based regimen, the lopinavir-based regimen had cHR 2.8, 95% CI 0.94 to 8.26. Independent risk factors for MetS with multivariable regression analysis were age ≥45 years (aHR 1.8, 95%

**Table 3** Associated risk factors for metabolic syndrome among HIV patients on cART at TASH and ZMH, Addis Ababa, Ethiopia

| | Metabolic syndrome* | | cHR (95%CI) | P value | aHR (95% CI) | P value |
|---|---|---|---|---|---|---|
| Variables | Yes | No | | | | |
| Age (years) | | | | | | |
| <45 | 80 (31.0%) | 178 (69.0%) | 1 | | 1 | |
| ≥45 | 113 (43.5) | 147 (56.5) | 1.68 (1.14 to 2.49) | 0.009 | 1.79 (1.18–2.37) | 0.006 |
| Sex | | | | | | |
| Male | 50 (29.4%) | 120 (70.6%) | 1 | | 1 | |
| Female | 143 (41.2%) | 204 (58.8%) | 1.71(1.19 to 2.45) | 0.003 | 1.79 (1.13–2.83) | 0.013 |
| Recent CD4 count (cells/mm$^3$) | | | | | | |
| <200 | 11 (29.0%) | 27 (71.0%) | 1 | | 1 | |
| 200–350 | 30 (29.4%) | 72 (70.6%) | 1.02 (0.45 to 2.32) | 0.957 | 0.99 (0.41–2.40) | 0.989 |
| >350 | 143 (41.3%) | 203 (59.7%) | 1.73 (0.83 to 3.60) | 0.143 | 1.50 (0.67–3.35) | 0.320 |
| BMI (kg/m$^2$) | | | | | | |
| ≤24.9 | 83 (27.0%) | 225 (73.0%9 | 1 | | 1 | |
| ≥25 | 108 (53.0%) | 6 (47.0%) | 3.05 (2.10 to 4.43) | 0.001 | 2.71 (1.81–4.06) | 0.001 |
| Current cART | | | | | | |
| NVP based | 24 (21.8%) | 86 (79.2%) | 1 | | 1 | |
| EFV based | 152 (44.1%) | 193 (55.9%) | 2.82 (1.71 to 4.65) | 0.001 | 2.79 (1.62–4.80) | 0.001 |
| LPV based | 7 (43.7%) | 9 (56.3%) | 2.79 (0.94 to 8.26) | 0.640 | 3.72 (1.04–13.3) | 0.043 |
| ATV based | 10 (21.3%) | 37 (78.7%) | 0.97 (0.42 to 2.23) | 0.940 | 1.03 (0.41–2.60) | 0.954 |

*Based on IDF 2006 Metabolic Syndrome Diagnostic Criteria.
ATV, atazanavir; BMI, body mass index; cART, combined antiretroviral treatment; cHR, crude hazard ratio; EFV, efavirenz; IDF, International Diabetes Federation; LPV, lopinavir; NVP, nevirapine; TASH, Tikur Anbessa Specialized Hospital; ZMH, Zewditu Memorial Hospital.

CI 1.2 to 2.4, p=0.006), female sex (aHR 1.8, 95% CI 1.1 to 2.8, p=0.013), BMI (kg/m$^2$) ≥25 (aHR 2.71, 95% CI (1.8 to 4.1), p=0.001), EFV-based cART regimen (aHR 2.8 95% CI 1.6 to 4.8, p=0.001) and LPV based cART regimen (aHR 3.7, 95% CI 1.0 to 13.3), p=0.043). CD4 count above 350 cells/mm$^3$ was not significantly associated with MetS in this study (cHR 1.7, 95% CI 0.8 to 3.6), p=0.143, aHR 1.5, 95% CI 0.67% to 3.35%), p=0.320) (table 3).

Associated risk factors for lipodystrophy on bivariate analysis were exposure to a stavudine containing regimen (cHR 3.1, 95% CI 1.7 to 5.7, p=0.001) and BMI (kg/m$^2$) ≥25 (cHR 2.8, 95% CI 1.8 to 4.2, p=0.001). Age above 45 years was not found to be associated with lipodystrophy in our study (p=0.567). An independent risk factor for lipodystrophy was a cART regimen containing stavudine (aHR 3.1, 95% CI 1.6 to 6.1, p=0.001). There was no significant association between CD4 count and recent cART regimen and lipodystrophy in this study (table 4).

## DISCUSSION

Our study revealed the prevalence of MetS is 37.3% by IDF 2006 criteria and 37.6% by ATP III 2005 criteria.[18 19] In contrast to an MetS prevalence in an HIV-negative study population with hypertension in Northern Ethiopia by Tachebele et al.[24] (40.7% IDF and 39.3% ATPIII), our

finding was slightly lower. The prevalence of MetS was higher with both criteria in our study than the 14% reported by Samaras et al,[25] (males 72.8%) by IDF criteria and 18% (males 84.9%) by ATPIII criteria. Berhane et al[12] reported from Southern Ethiopia a prevalence of 21.1% by IDF criteria. The prevalence of MetS in our setting is higher than that reported in a Tanzanian study which revealed a 25.6% prevalence using IDF criteria.[26] This study's finding of the prevalence of MetS is lower than an Italian study that reported 45.4% prevalence by ATPIII criteria.[27] Compared with our study finding, the prevalence of MetS using IDF criteria was 4.8% (8.6% in females and 1.8% in males) in the national NCD STEPS survey in the Ethiopian general population.[28] Undoubtedly, our finding indicates a significantly higher risk of CVD among PWH receiving cART. The above discrepancies in the prevalence rates are partly explained by the time differences between the studies, the types of cART taken and the populations studied. Even though the IDF criteria appears to be more stringent because of the requirement of waist circumference, it also allows more patients with lesser waist circumference to be included as was evident in this result indicating a closer prevalence to the ATP III criteria. Similarly, Samaras et al[25] have found a higher prevalence of MetS by ATPIII criteria. A

**Table 4** Associated risk factors for lipodystrophy among HIV patients on cART at TASH and ZMH, Addis Ababa, Ethiopia

| | Lipodystrophy sign(s) | | cHR (95% CI) | P value | aHR (95% CI) | P value |
|---|---|---|---|---|---|---|
| Variables | Yes | No | | | | |
| Initial cART* | | | | | | |
| D4T exposed | 36 (52.2%) | 33 (47.8%) | 3.09 (1.68 to 5.68) | 0.001 | 3.13 (1.6 to 6.12) | 0.001 |
| D4T not exposed | 35 (26.1%) | 99 (73.9%) | 1 | | 1 | |
| CD4 count (cells/mm$^3$) | | | | | | |
| <200 | 6 (16.7%) | 30 (83.3%) | 1 | | 1 | |
| 200–350 | 18 (17.3%) | 86 (82.7%) | 1.14 (0.42 to 3.14) | 0.795 | 0.63 (0.18 to 2.18) | 0.462 |
| >350 | 92 (26.6%) | 254 (73.4%) | 1.93 (0.78 to 4.77) | 0.153 | 0.62 (0.27 to 1.41) | 0.254 |
| BMI (kg/m$^2$) | | | | | | |
| ≤24.9 | 71 (34.7%) | 133 (65.2%) | 1 | | 1 | |
| ≥25 | 50 (16.2%) | 258 (83.8%) | 2.76 (1.81 to 4.18) | 0.001 | 1.48 (0.76 to 2.86) | 0.249 |
| Current cART | | | | | | |
| NVP based | 24 (21.8%) | 86 (78.2%) | 1 | | 1 | |
| EFV based | 85 (24.6%) | 260 (75.4%) | 1.17 (0.70 to 1.96) | 0.547 | 1.20 (0.52 to 2.78) | 0.666 |
| LPV based | 0 | 16 (100%) | 0.00 (00-) | 0.998 | 0.00 (0.00-) | 0.999 |
| ATV based | 13 (27.7%) | 34 (72.3%) | 1.37 (0.63 to 3.00) | 0.431 | 0.90 (0.29 to 2.77) | 0.857 |

*Intial cART regimen: for those participants whose drugs were changed to the current cART regimen; D4T: stavudine.
ATV, atazanavir; BMI, body mass index; cART, combined antiretroviral therapy; LPV, lopinavir; NVP, nevirapine; TASH, Tikur Anbessa Specialized Hospital; ZMH, Zewditu Memorial Hospital.

slightly higher value of MetS prevalence with IDF criteria was reported by Hirigo and Tesfaye in Southern Ethiopia, 24.3% and 17.8% using the IDF and NCEP-ATP III criteria, respectively.[29]

Independent predictors of MetS in our study were age above 45 years, female sex, BMI (kg/m$^2$) above 25, EFV-based cART regimen and LPV-based cART regimen. Similar to our findings, studies in Ethiopia and Tanzania have found a significant association between female sex and MetS.[26 29] This is in line with other reports.[25 30] In contrast, one study revealed that gender has no significant association with MetS.[31] Similar to our study findings, Hirigo and Tesfaye have reported an increasing age above 40 years to be significantly associated with MetS.[29] This is in line with the report from Spain that has shown an association between increasing age and MetS.[25] Similar to our study finding, higher BMI was significantly associated with MetS in other studies.[25 29] Furthermore, several other studies illustrated that the BMI is a quantitative predictor of MetS.[32–34]

Current use of EFV-based regimens is significantly associated with MetS in our study. This is supported by the fact that non-nucleoside reverse transcriptase inhibitors (NNRTIs) impair lipid metabolism as a class,[35] and efavirenz leads to increased inflammation,[36 37] impaired adipogenesis and adipocyte differentiation,[36] impaired metabolism[36 37] and no mitochondrial toxicity.[37] Efavirenz use is associated with increased apoptosis, increased mitochondrial mass and oxidative stress in hepatic cells. Disruption in mitochondrial membrane potential was observed in hepatocytes promoting cytochrome c release

and apoptosis.[38 39] Efavirenz also inhibits complex I of the electron transport chain stimulating ROS production and a decrease in ATP which leads to an increase in the lipid content of hepatic cells.[40] Efavirenz also caused increased mitochondrial depolarisation and altered mitochondrial morphology leading to mitophagy (clearance of damaged mitochondria) in neuronal cells.[41] Mitochondrial dysfunction promotes the progression of MetS by encouraging the occurrence of inflammation and oxidative stress. Efavirenz also acts as a pregnane×receptor agonist that promotes dyslipidaemia and hypercholesteraemia.[42] NNRTIs promote MetS by reducing insulin sensitivity through their proinflammatory effects. NNRTIs are known for their reduction of adiponectin concentrations through increased expression of TNF-a, IL-6 and IL-1B. The reduction in the insulin sensitivity modulator, adiponectin, allows for the progression of insulin resistance.[43] Current EFV treatment has been correlated with elevated markers of atherosclerosis.[44] The finding of liponavir-based cART regimen is in line with the independent association with MetS in a study in Barcelona, Spain.[25] This is supported by the molecular mechanism of the drug that causes substantial impairment of adipocyte differentiation and lipid or glucose metabolism, pronounced decrease in mitochondrial proteins and increased inflammation.[45]

The prevalence of lipodystrophy in this study was found to be 22.6% by subjective participants report and 23.6% by objective assessment for lipodystrophy manifestations. Our study finding is higher than the report of 12.1% by Berhane et al[12] from South West Ethiopia and lower than the 68.3%

prevalence in the study from Addis Ababa.[13] In our study, the higher prevalence compared with Berhane *et al*[12] finding might be attributable to the larger sample size and the lower prevalence compared with Feleke *et al*[13] finding may be explained by the trends of cART drug prescription that has been towards withdrawal of stavudine. In our study, the prevalence of hypertriglyceridaemia $\geq 150\,mg/dL$, a marker of lipodystrophy with insulin resistance and a component of MetS was found in 40.7% of participants. This finding is closer to the finding of 43.5% prevalence in urban Tanzanian HIV population on cART,[26] higher than the 15.2% prevalence finding in Addis Ababa,[13] and lower than the report of 59.6% prevalence in a study in defence hospital in Addis Ababa.[46] Unadjusted, Lipodystrophy was significantly associated with a stavudine-containing regimen and BMI $(kg/m^2)$ $\geq 25$. However, after adjustment for CD4 count $(cells/mm^3)$ and current cART, only D4T exposure was significantly associated with lipodystrophy. This finding of strong association between D4T and lipodystrophy is in line with previous reports.[47] This is because of the nucleoside reverse transcriptase class effect on impairing lipid metabolism,[35] and D4T effect of reduction of mitochondrial DNA.[48 49] Caron *et al*[50] in their in vitro study have shown that stavudin leads to decreases in lipid content, survival of adipose cells and mitochondrial activity. The amount of mtDNA was found to be significantly decreased among patients receiving NRTIs indicating the mechanistic effect of stavudin on mitochondrial toxicity.[51] The resultant inflammatory cytokines such as interleukins results in adipose cell destruction. This is known to impair Insulin signalling function and glucose transport. ART-associated lipodystrophy increases the risk for dyslipidaemia, insulin resistance, diabetes mellitus and heart disease.[52]

Limitations of this study are the cross-sectional design, difficulty to trace some of the records on the initial cART regimen from old records, difficulty to get adequate samples for some of the variables that may have influenced the outcome like patients on atazanavir-based regimen, and fewer patients who were smokers or taking alcohol to identify their potential correlation in the study participants, and fewer patients on liponavir-based regimen as well. In addition, a complete nutritional assessment was not made. Advanced techniques for the assessment of lipodystrophy were not used because of availability. Of all the studies done in Ethiopia on MetS and Lipodystrophy, this is the one with the largest sample size and laboratory test results available for all participants.

## CONCLUSION

MetS was diagnosed in 37.3% and 37.6% of PWH using IDF and ATPIII criteria, respectively. The higher prevalence of MetS is associated with female sex, age above 45 years, BMI above $25\,kg/m^2$, EFV-based and Liponavir-based cART regimens. Hypertriglyceridaemia was common among PWH receiving cART indicating cardiovascular and metabolic risks. Lipodystrophy was detected in 23.6% of study participants. Previous stavudine exposure is significantly associated with lipodystrophy. These metabolic changes among those on cART place them at high risk of cardiovascular events. Therefore, screening for MetS in PWH receiving cART and managing modifiable CV risk factors are highly recommended. Furthermore, a prospective cohort study is recommended to evaluate MetS and lipodystrophy and the related CV outcomes among patients on cART.

**Acknowledgements** The authors would like to acknowledge the study participants for their voluntary participation, the data collectors and the study centres staff at Zewditu Memorial Hospital and Tikur Anbessa Specialized Referral Hospital, for their invaluable support in the conduct of our study. In addition, the authors acknowledge Ms. Dorqa Woldesenbet (MPH) from School of Public Health-Addis Ababa University for her support in the statistical analysis.

**Contributors** AAA, IO and WAD conceptualised the study. AAA collected the primary data, conducted the analysis and drafted the manuscript. HY, AR, AS, IO and WAD contributed to conceptualisation, collection of data and data analysis. AAA, HY, AR, AS, IO and WAD have read, reviewed and approved the article. AAA and WAD are the authors acting as guarantor.

**Funding** This collaborative research work is financially supported by an NIH funded project, Medical Education Partnership Initiative-Junior Faculty (D43TW010143).

**Competing interests** None declared.

**Patient and public involvement** Patients and/or the public were not involved in the design, or conduct, or reporting, or dissemination plans of this research.

**Patient consent for publication** Not applicable.

**Ethics approval** The study has been reviewed and approved by the Institution Review Board of the College of Health Sciences-Addis Ababa University with the Ref/ID# 077/17/IM and by Addis Ababa city regional health bureau ethics review committee and has therefore been performed in accordance with the ethical standards of the Declaration of Helsinki. Information sheet detailing the procedure of the study, the benefits the of the study, the full right to agree or disagree to participate in the study, assurance of confidentiality, continuation of their usual medical care regardless of their agreement or disagreement to participate in the study, and the linkage to the treating team if there are clinical or lab findings that require attention was shared to study participants. For participants who cannot read or had questions, data collectors had read, responded and explained. Written informed consent was obtained from all participants of the study.

**Provenance and peer review** Not commissioned; externally peer reviewed.

**Data availability statement** Data are available on reasonable request.

**ORCID iD**
Abdurezak Ahmed Abdela http://orcid.org/0000-0001-7795-752X

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
