## [Reviewer comments · BMJ Open]

ARTICLE DETAILS

TITLE (PROVISIONAL)	THE PREVALENCE AND RISK FACTORS OF METABOLIC SYNDROME IN ETHIOPIA: DESCRIBING AN EMERGING OUTBREAK IN HIV CLINICS OF THE SUB-SAHARAN AFRICA. A CROSS SECTIONAL STUDY
AUTHORS	Abdela, Abdurezak; Yifter, Helen; Reja, Ahmed; Shewaamare, Aster; Ofotokun, Ighovwerha; Degu, Wondwossen

VERSION 1 – REVIEW

REVIEWER	Lima, Klealdo Federal University of Pernambuco
REVIEW RETURNED	10-Jan-2023

GENERAL COMMENTS	The article reveals information about the prevalence and risk factors associated with metabolic syndrome and lipodystrophy, mainly regarding antiretroviral therapy, in Ethiopia. The text is very informative on the subject, despite analyzing antiretroviral therapies that are in disuse in many countries, for example, it did not evaluate integrase inhibitors. However, it reveals the local reality regarding the epidemic and the treatments adopted. The results obtained are expressive, however, I missed the collection of clinical data, for example, the presence of co-infections such as HBV, HCV and syphilis. ABSTRACT 1- Its methods section only specified the study site and statistical methods. The study population and studied variables were not described. 2- In the results section: - What do the authors define as the Median recent CD4 count? INTRODUCTION 1- In the sentence: "There was an estimated 36.7 million individuals living with HIV infection by 2015." Could the authors cite a more up-to-date reference? METHODS 1- "Study participants were included if they were above the ages of 18 years, had documented HIV infection and adhering to and taking cART for at least 12 months". How was cART adherence measured? 2- PATIENT AND PUBLIC INVOLVEMENT: None. Why include this topic? 3- "Body fat distribution was assessed by patient's self-report and physical examination". Generally, because it is a specific measure, this variable is not indicated to be evaluated through the patient's self-report.
--

	4- "Mendray chemistry machine". It is recommended to inform the manufacturer of the automated device. 5- "MetS was defined using International Diabetes Federation(IDF) (17) and the National Cholesterol Education Program's Adult Treatment Panel (ATP III)(18) criteria." Were both criteria used for the diagnosis of MetS? Would only one criterion be necessary, as the guidelines have different criteria? 6- I would like to know if this research has been approved by the National or Regional Ethics Committee? RESULTS 1- At the bottom of table 1, Zidovudine is spelt incorrectly: "Ziduvidin (ZDV)" 2- "Distribution of Lipodystrophy symptoms were loss of buccal fat pad 76(14.7%), central adiposity \pmextremity wasting 80(15.4%), dorsocervical/supraclavicular fat pad 39(7.5%), and breast enlargement 31(6.0%). Signs of Lipodystrophy were noted as loss of buccal fat pad 60 (11.6%), central adiposity \pmextremity wasting 61(11.8%), dorsocervical/supraclavicular fat pad 1(0.2%).(Table 2)". What is the difference between the signs and symptoms of lipodystrophy? This measure and difference were not mentioned in the methodology. 3- Particularly, I found table 3 redundant, as all the information contained therein is in the text. DISCUSSION 1- Regarding the DISCUSSION section, the authors should report, with more scientific audacity, hypotheses related to the association of efavirenz and stavudine in the metabolic syndrome and lipodystrophy. There was greater emphasis, in the discussion, on debating the prevalence and risk factors of these events than what was discussed above.
--	---

REVIEWER	Masenga, Sepiso Mulungushi University, School of medicine and Health sciences
REVIEW RETURNED	27-Apr-2023

GENERAL COMMENTS	This study addresses the prevalence and correlates of Metabolic syndrome among persons with HIV from Ethiopia. The study is written well. I have a few suggestions:  1. I did not understand the section "Patient and public involvement : None" 2. Not sure meaning of "Body fat distribution was assessed by patient's self-report .." on page 8. What exactly did the patient report on fat distribution. This should be clarified to differentiate physical examination findings by the doctor and patient report. 3. Table 1: Better to put the drugs in the table rather than on footnote 4. Tables 1 - 3 can be combined 5. What was the rationale for categorizing age in logistic regression
---

	6. Some variables not indicated when analyzing risk factors for MetS and Lipodystrophy in logistic regression. e.g. age not included in the model 7. The study is focused on both MetS and Lipodystrophy but the title only indicates MetS?
--	---

VERSION 1 – AUTHOR RESPONSE

This study addresses the prevalence and correlates of Metabolic syndrome among persons with HIV from Ethiopia. The study is written well. I have a few suggestions:

1. I did not understand the section "Patient and public involvement : None"
2. Not sure meaning of "Body fat distribution was assessed by patient's self-report .." on page 8. What exactly did the patient report on fat distribution. This should be clarified to differentiate physical examination findings by the doctor and patient report.
3. Table 1: Better to put the drugs in the table rather than on footnote
4. Tables 1 - 3 can be combined
5. What was the rationale for categorizing age in logistic regression
6. Some variables not indicated when analyzing risk factors for MetS and Lipodystrophy in logistic regression. e.g. age not included in the model
7. The study is focused on both MetS and Lipodystrophy but the title only indicates MetS?

Authors response:

The title is to highlight the rising Metabolic Syndrome in the HIV patients that have conglomerates of atherosclerotic cardiovascular disease risks. MetS components have been studied supported by lab data in the study population. As well, lipodystrophy by itself through inducing inflammation can be a risk factor for metabolic syndrome.

VERSION 2 – REVIEW

REVIEWER	Lima, Kledaldo Federal University of Pernambuco
REVIEW RETURNED	21-Aug-2023

GENERAL COMMENTS	The paper is a description of prevalence and risk factors of MetS in Ethiopia, in HIV-infected Ethiopian taking cART. the text is a response to reviewers and has demonstrated a major upgrade compared to the previously reviewed text. However, we have a few notes: Page 37 line 12: "the data collection of adverse events of anti-HIV drugs..." I would suggest detailing the period for obtaining these data.
--

	Page 39 line 49 why were patients with diabetes mellitus and hypertension excluded? Has this inclusion not imposed an underestimation of the MetS/ Ethics considerations patients did not sign an informed consent form? Page 53 line 31: the prediction of risk factors for MetS was performed after adjusting for CD4. But, in the results there is no such explanation. Another consideration, why was there an adjustment for CD4? The References section should be revised and updated. Some references have the information: "this article on pubmed", unnecessary! Other references are not correctly described (example: references 15, 28, 44, 47). There are references with doi and others without doi. Web site references must be dated.
--	---

REVIEWER	Masenga, Sepiso Mulungushi University, School of medicine and Health sciences
REVIEW RETURNED	07-Aug-2023

GENERAL COMMENTS	The authors have addressed all my concerns. I have no further comments.
---

VERSION 2 – AUTHOR RESPONSE

Reviewer: 2

Dr. Sepiso Masenga, Mulungushi University

Comments to the Author:

The authors have addressed all my concerns. I have no further comments.

Authors response:

Thank you.

Reviewer: 1

Dr. Kleodoaldo Lima, Federal University of Pernambuco

Comments to the Author:

The paper is a description of prevalence and risk factors of MetS in Ethiopia, in HIV-infected Ethiopian taking cART.

the text is a response to reviewers and has demonstrated a major upgrade compared to the previously reviewed text. However, we have a few notes:

Page 37 line 12:

"the data collection of adverse events of anti-HIV drugs..."

I would suggest detailing the period for obtaining these data.

Authors response:

We have added:

" Though this analysis was made for data until 2008, DAD data were obtained in cohorts I-III in the period of December 1999-2001, then throughout spring of 2004, and until 2010 respectively. "

Page 39 line 49

why were patients with diabetes mellitus and hypertension excluded? Has this inclusion not imposed an underestimation of the MetS/

Authors response:

Patients with diagnosis of diabetes and hypertension before starting cART were excluded to avoid overestimation of MetS as we are looking into the prevalence of MetS after they are started on cART.

Ethics considerations

patients did not sign an informed consent form?

Authors response:

All included patients have signed the informed consent.

We have added this in the ethical considerations section.

Page 53 line 31:

the prediction of risk factors for MetS was performed after adjusting for CD4. But, in the results there is no such explanation. Another consideration, why was there an adjustment for CD4?

Authors response:

Thank you for the constructive comment. We agree with this comment. No such adjustment was done and it is not needed. We have corrected accordingly.

The References section should be revised and updated. Some references have the information: "this article on pubmed", unnecessary!

Other references are not correctly described (example: references 15, 28, 44, 47).

There are references with doi and others without doi.

Web site references must be dated.

Authors response:

Thank you for the comment. We have updated all the references, added doi where available, and dated web site references.

Reviewer: 2

Competing interests of Reviewer: none

Reviewer: 1

Competing interests of Reviewer: None.